# Factors Associated with Financial Security, Food Security and Quality of Daily Lives of Residents in Nigeria during the First Wave of the COVID-19 Pandemic

**DOI:** 10.3390/ijerph18157925

**Published:** 2021-07-27

**Authors:** Morenike Oluwatoyin Folayan, Olanrewaju Ibigbami, Maha El Tantawi, Brandon Brown, Nourhan M. Aly, Oliver Ezechi, Giuliana Florencia Abeldaño, Eshrat Ara, Martin Amogre Ayanore, Passent Ellakany, Balgis Gaffar, Nuraldeen Maher Al-Khanati, Ifeoma Idigbe, Anthonia Omotola Ishabiyi, Mohammed Jafer, Abeedha Tu-Allah Khan, Zumama Khalid, Folake Barakat Lawal, Joanne Lusher, Ntombifuthi P. Nzimande, Bamidele Emmanuel Osamika, Mir Faeq Ali Quadri, Mark Roque, Ala’a B. Al-Tammemi, Muhammad Abrar Yousaf, Jorma I. Virtanen, Roberto Ariel Abeldaño Zuñiga, Joseph Chukwudi Okeibunor, Annie Lu Nguyen

**Affiliations:** 1Mental Health and Wellness Study Group, Obafemi Awolowo University, Ile-Ife 220282, Nigeria; oibigbami@oauife.edu.ng (O.I.); maha_tantawy@hotmail.com (M.E.T.); brandon.brown@medsch.ucr.edu (B.B.); nourhanovic@gmail.com (N.M.A.); oezechi@yahoo.co.uk (O.E.); gflorabeldano@gmail.com (G.F.A.); eshrataslam@gmail.com (E.A.); mayanore@uhas.edu.gh (M.A.A.); pellakany@iau.edu.sa (P.E.); bgosman@iau.edu.sa (B.G.); nuraldeen.alkhanati@gmail.com (N.M.A.-K.); ifeomagenia@yahoo.com (I.I.); paduatonia@yahoo.com (A.O.I.); dr.mjafer@gmail.com (M.J.); abeedha.sbs@pu.edu.pk (A.T.-A.K.); zumama.mphil.sbs@pu.edu.pk (Z.K.); folakemilawal@yahoo.com (F.B.L.); joanne.lusher@uws.ac.uk (J.L.); nzimandentombifuthi@gmail.com (N.P.N.); osamika.bamidele@lcu.edu.ng (B.E.O.); dr.faeq.quadri@gmail.com (M.F.A.Q.); markyroquedpa@gmail.com (M.R.); alaa.tammemi@med.unideb.hu (A.B.A.-T.); abrar.ms.zool@pu.edu.pk (M.A.Y.); jorma.virtanen@uib.no (J.I.V.); ariabeldanho@gmail.com (R.A.A.Z.); okeibunorj@who.int (J.C.O.); annie.nguyen@med.usc.edu (A.L.N.); 2Department of Child Dental Health, Obafemi Awolowo University, Ile-Ife 220282, Nigeria; 3Department of Mental Health, Obafemi Awolowo University, Ile-Ife 220282, Nigeria; 4Department of Pediatric Dentistry and Dental Public Health, Faculty of Dentistry, Alexandria University, Alexandria 21527, Egypt; 5Department of Social Medicine, Population and Public Health, School of Medicine, University of California, Riverside, CA 92501, USA; 6Department of Clinical Sciences, Nigerian Institute of Medical Research, Lagos 101212, Nigeria; 7Institute for Research on Public Health, School of Medicine, University of Sierra Sur, Oaxaca 70805, Mexico; 8Government College for Women, Moulana Azad Road, Srinagar Kashmir, J&K 190001, India; 9Department of Health Policy Planning and Management, University of Health and Allied Sciences, PMB 31 Ho, Ghana; 10Department of Substitutive Dental Sciences, College of Dentistry, Imam Abdulrahman Bin Faisal University, Dammam 31441, Saudi Arabia; 11Department of Preventive Dentistry, College of Dentistry, Imam Abdulrahman Bin Faisal University, Dammam 31441, Saudi Arabia; 12Department of Oral and Maxillofacial Surgery, Faculty of Dentistry, Syrian Private University, Damascus 36822, Syria; 13Centre for Rural Health, School of Nursing and Public Health, University of KwaZulu-Natal, Durban 4001, South Africa; 14Department of Preventive Dental Sciences, Faculty of Dentistry, Jazan University, Jazan 45142, Saudi Arabia; 15Department of Health Promotion, Faculty of Health, Medicine and Life Sciences, Maastricht University, 6200 MD Maastricht, The Netherlands; 16School of Biological Sciences, University of the Punjab, Quaid-i-Azam Campus, Lahore 54590, Pakistan; 17Department of Periodontology and Community Dentistry, University of Ibadan, Ibadan 200212, Nigeria; 18School of Health and Life Sciences, University of the West of Scotland, London E142BE, UK; 19Department of Economic and Human Geography, Faculty of Geosciences, University of Szeged, H-6722 Szeged, Hungary; 20Department of Psychology, Faculty of Environment, Management and Social Sciences, Lead City University, Ibadan 200255, Nigeria; 21Division of Dental Public Health, Department of Preventive Dentistry, Jazan University, Jazan 45142, Saudi Arabia; 22Department of Maternity & Childhood Nursing, College of Nursing, Taibah University, Madinah 42223, Saudi Arabia; 23Department of Family and Occupational Medicine, Faculty of Medicine, Doctoral School of Health Sciences, University of Debrecen, H-4032 Debrecen, Hungary; 24Institute of Zoology, University of the Punjab, Quaid-i-Azam Campus, Lahore 54590, Pakistan; 25Faculty of Medicine, University of Turku, FI-20014 Turku, Finland; 26Post Graduate School, University of Sierra Sur, Oaxaca 70800, Mexico; 27Research Development and Innovations, Assistant Regional Director Cluster, WHO Regional Office for Africa, Brazzaville BP 06, Congo; 28Department of Family Medicine, Keck School of Medicine, University of Southern California, Los Angeles, CA 91803, USA

**Keywords:** SARS-CoV-2, economic security, depression, anxiety, financial security, pandemic, HIV, AIDS, Nigeria

## Abstract

An online survey was conducted to identify factors associated with financial insecurity, food insecurity and poor quality of daily lives of adults in Nigeria during the first wave of the COVID-19 pandemic. The associations between the outcome (experience of financial loss, changes in food intake and impact of the pandemic on daily lives) and the explanatory (age, sex, education level, anxiety, depression, HIV status) variables were determined using logistic regression analysis. Of the 4439 respondents, 2487 (56.0%) were financially insecure, 907 (20.4%) decreased food intake and 4029 (90.8%) had their daily life negatively impacted. Males (AOR:0.84), people who felt depressed (AOR:0.62) and people living with HIV -PLHIV- (AOR:0.70) had significantly lower odds of financial insecurity. Older respondents (AOR:1.01) had significantly higher odds of financial insecurity. Those depressed (AOR:0.62) and PLHIV (AOR:0.55) had significantly lower odds of reporting decreased food intake. Respondents who felt anxious (AOR:0.07), depressed (AOR: 0.48) and who were PLHIV (AOR:0.68) had significantly lower odds of reporting a negative impact of the pandemic on their daily lives. We concluded the study findings may reflect a complex relationship between financial insecurity, food insecurity, poor quality of life, mental health, and socioeconomic status of adults living in Nigeria during the COVID-19 pandemic.

## 1. Introduction

The impact of COVID-19 pandemic varied between regions and countries. Food insecurity was worse in the most fragile regions of the world [1]. There were also substantial variations in anxiety and depression symptoms across countries during the COVID-19 lockdown [2,3]. In many countries around the world, many organisations and businesses had to shut down to maximise public safety resulting in significant loss of revenue. As a result, staff were laid off, work hours were reduced, salaries were cut, and furloughs were enacted in a bid to meet organisations’ operating costs [4,5]. These types of market shocks have unequal consequences on the financial well-being of populations as a whole [6] with the impact varying between populations in-country thereby worsening the existing inequalities [7,8]. Women, young people, and adults aged 65 and older were more likely to be financially vulnerable during the pandemic while household heads with at least a tertiary education have a lower probability of being affected [9]. Working-age households may also face financial vulnerabilities and intersecting health, financial and other socioeconomic vulnerabilities [10].

It is likely that the impact of the economic shock caused by the pandemic will differ between populations [7]. For example, people living with HIV (PLHIV) may be more vulnerable to the COVID-19 pandemic’s economic impact compared to those not living with HIV. This is because a high proportion of PLHIV work in industries that are more prone to COVID-19 disruptions; and PLHIV have disproportionately higher unemployment and food insecurity rates relative to the general population [11,12]. Among PLHIV in both low- and high-resource settings, food insecurity is more prevalent and well above the general population estimates, with women being most at risk [13]. Unemployment, drug misuse and worse physical and mental health status have also been associated with food insecurity for PLHIV [14]. Also, in times of economic crisis where jobs are not in abundance, PLHIV may be excluded from the labour market because of poor health status [15].

The economic shock caused by the pandemic may also affect men and women differently. Women’s jobs are 1.8 times more vulnerable to the impact of the pandemic than men’s jobs [16]. Women make up 39% of global employment but account for 54% of overall job losses [16] with concerns that the pandemic may widen the existing gender inequality in job placements [17,18]. There also may be age, income and educational status differences in the impact of the pandemic. Younger populations in different countries are affected more though the impact across income groups is less clear and less consistent across countries [19]. However, there are few accessible publications on the disparity in the COVID-19 pandemic by educational status.

In addition to the negative impacts on financial and food security, the daily lives of many were disrupted with far reaching consequences for some. These impacts include the disruption of the medical supply chain and neglect of some health problems due to COVID-19 management prioritisation [20]. Schools, social events, travel, leisure, and entertainment have also been disrupted [21]. This impact had mental health consequences like anxiety and depression [22] that may be worse for people living in sub-Saharan Africa. Access to mental health care through telemedicine services may be challenging due to the low digital literacy, low smartphone penetration and limited internet connection which may make the provision of online mental health services a limited option in developing countries [23]. Depression and anxiety may also affect the use of telemedicine services thereby affecting daily lives due to poor access to healthcare during the pandemic [24].

Few studies have examined the impact of the COVID-19 pandemic on financial and food security in sub-Saharan Africa. Yet, sub-Saharan Africa is home to a many people living in poverty [25] who are likely to be negatively affected by the economic impact of the pandemic. The region is also home to many people with poor health. For example, the region is host to 67% of the population of PLHIV [26] and Nigeria is home to the second largest population of PLHIV [27]. Nigeria was also ranked 100 out of 113 nations in terms of food security [28]; and the country has a poor financial security profile. Lack of job opportunities is at the core of high poverty levels [29]. More than 80% of the population is employed in the informal economic sector and thus, dependent on daily income [30,31]. Between March and June 2020, the country instituted eight weeks of lockdown to control the pandemic and the total gross domestic product fell by 23% during that time [30]. The pandemic resulted in about 7 million newly poor people in 2020 [28]. In addition, the agri-food system gross domestic product also fell by 11% primarily due to restrictions on food services. Household incomes also fell by a quarter leading to a 9% point increase in the national poverty rate [30]. Understanding the impact of the pandemic on the financial vulnerability, food security and daily lives of people in countries in sub-Saharan Africa like Nigeria, may help policy makers to modify their programs and interventions to improve the current COVID-19 response by reaching out to persons most in need of food and financial support.

The aim of the present study was to identify factors associated with financial and food insecurity; as well as factors that affected the daily life of people in Nigeria during the first wave of the COVID-19 pandemic. Specifically, the study identified factors associated with financial loss and a decrease in food intake because of the COVID-19 pandemic. It was hypothesized that factors associated with financial loss and food insecurity can also be associated with the negative impact of the COVID-19 pandemic on the daily life of adults living in Nigeria.

## 2. Materials and Methods

The data for this analysis was extracted from a cross-sectional multi-country survey. The international survey solicited information about mental health and wellness from a global convenience sample of adults aged 18 years and above from July to December 2020. Data were collected using an online survey platform (Survey Monkey, Momentive Inc.: San Mateo, CA, USA). Study participants for the global survey were recruited through respondent-driven sampling. Initial participants reached by the 45 data collectors, were asked to share the survey link with their contacts within their countries. The survey links were also posted on social media groups (Facebook, Twitter, and Instagram), network email lists and WhatsApp groups. Ethical approval for the study was obtained from the Human Research Ethics Committee at the Institute of Public Health of the Obafemi Awolowo University Ile-Ife, Nigeria (HREC No: IPHOAU/12/1557).

The data collection tool was initially developed for a study targeting a specific population in the United States and was consequently adapted and validated for global use [32]. The questionnaire took an average of 11 min to be completed and was administered in English. Study participants were asked to complete an anonymous, closed-ended questionnaire about how COVID-19 affected their wellness and well-being. The questionnaire enquired about the sociodemographic profile, COVID-19 impact on food intake and security, financial security and their daily life. Data of participants from Nigeria were extracted for the purpose of this study.

### 2.1. Explanatory Variables

Sociodemographic variables: The section on sociodemographic profile collected data on age, sex at birth and highest level of education attained (none, primary, secondary, college/university).

Mental health status: The questions were part of the Pandemic Stress Index assessing the psychosocial impact of COVID-19 [33]. The mental health status of the respondents was assessed by asking respondents to select from eight feelings that they may have experienced during COVID-19. Only two of the eight feelings assessed were used for this paper namely: anxiety and depression. Each were treated as dichotomous variables (yes/no).

HIV status: A question was also asked about HIV status. Respondents self-identified their HIV status as positive, negative, unknown, or indicated if they were unwilling to declare.

### 2.2. Outcome Variables

Impact of pandemic on financial security: Respondents were asked if they experienced a financial loss because of the COVID-19 pandemic and possible responses included, “yes”, “no” or “I don’t know”. The question was adopted from the Multi-Center AIDS Cohort Study [34].

Impact of pandemic on food intake: The question enquiring about food security was adopted from the Pandemic Stress Index [33]. Respondents were asked if their food intake had changed during the pandemic. The possible responses were: “increased”, “decreased”, or “no change”.

Impact of pandemic on daily lives: The impact of the pandemic on the daily lives of respondents was assessed by a single question. Respondents were asked: How much is/did COVID-19 negatively impact your day-to-day life? The response options were “extremely”, “very much”, “much”, “a little” and “not at all”. The responses were dichotomised to “had negative impact” (extremely, very much, much, a little) and “no negative impact” (not at all) [32].

### 2.3. Data Analysis

We performed multiple best-practice procedures to ensure the quality of our web-based survey data [35]. Each participant could only complete a single questionnaire through intellectual property address restrictions, though they could edit their answers freely until they choose to submit. The responses were downloaded from Survey Monkey as an SPSS Version 23.0 file (IBM SPSS Statistics for Windows, Armonk, NY: IBM Corp), cleaned and use for analysis. We identified and removed responses completed below seven minutes (*n* = 77); and those with incomplete data with respect to financial and food security questions and the question on COVID-19 impact on daily life (*n* = 252).

Descriptive analysis of the study variables was conducted. The test of associations between the outcome variables (impact of pandemic on financial security, food intake and daily lives) and the explanatory variables (age, sex, educational status, HIV status and mental health status), as well as with each other, were conducted using chi-square test (and *t*-test for age). Three logistic regression models were developed to identify the associations between the explanatory variables and each of the three outcome variables. The variables with a *p* value of 0.5 or less were included in the models. Adjusted odds ratios (AORs) and 95% confidence intervals (CIs) were calculated. The model fitness was assessed using the Nagelkerke R2, the Hosmer Lemeshow goodness of fit test and the Omnibus test of model coefficients. Statistical significance was set at 5%.

## 3. Results

There were 4439 respondents from Nigeria. Their mean age was 38.34 (SD = 11.6) years, ranging from 18 to 85 years. The majority of respondents were females (53.2%) and had tertiary education (80.9%). There were 746 (16.8%) respondents who reported being anxious, 389 (8.8%) felt depressed, 2487 (56.0%) reported having financial insecurity, 907 (20.4%) reported having a decrease in food intake during the pandemic and 4029 (90.8%) had their daily lives negatively impacted by the pandemic. Also, 912 (20.5%) respondents self-reported living with HIV.

As shown in Table 1, more respondents who had financial insecurity were older (*p* < 0.001), had no formal education, (*p* = 0.028), felt anxious (*p* = 0.041), depressed (*p* < 0.001) and were PLHIV (*p* < 0.001). Also, more respondents who reported a decrease in food intake had primary education (*p* < 0.001), felt anxious (*p* < 0.001), depressed (*p* < 0.001) and were PLHIV (*p* < 0.001). Similarly, more respondents who reported a negative impact of the pandemic on their daily life had primary school education (*p* = 0.002), felt anxious (*p* < 0.001), depressed (*p* < 0.001) and were PLHIV (*p* = 0.006).

As shown in Table 2, respondents who were male (AOR: 0.84; 95% CI: 0.75–0.95; *p* = 0.005), who felt depressed (AOR: 0.62; 95% CI: 0.50–0.79; *p* < 0.001) and PLHIV (AOR: 0.70; 95% CI: 0.59–0.83; *p* < 0.001) and had significantly lower odds of reporting financial insecurity. On the other hand, older respondents (AOR: 1.01; 95% CI: 1.00–1.02; *p* < 0.001) had significantly higher odds of reporting financial insecurity.

Respondents who felt depressed (AOR: 0.62; 95% 0.48–0.78; *p* < 0.001) and who were PLHIV (AOR: 0.55; 95% CI: 0.46–0.67; *p* < 0.001) had significantly lower odds of reporting decreased food intake.

Also, respondents who felt anxious (AOR: 0.08; 95% CI: 0.03–0.17; *p* < 0.001), depressed (AOR: 0.48; 95% CI: 0.27–0.85; *p* = 0.012) and who were PLHIV (AOR: 0.68; 95% CI: 0.50–0.92; *p* = 0.012) had significantly lower odds of reporting a negative impact of the pandemic on their daily life.

## 4. Discussion

The study findings underscore the complex relationships between factors associated factors associated with financial insecurity, decrease in food intake and the negative impact of COVID-19 among adults in Nigeria. First, the two factors that were associated with the three outcome variables were feeling depressed and living with HIV. Both factors reduced the odds of experiencing the financial insecurity, decrease in food intake and the negative impact of COVID-19 on the daily life of adults in Nigeria. Second, respondents who felt anxious had lower odds of reporting a negative impact of the epidemic on their daily lives. Third, socio-demographic variables were only associated with financial insecurity: financial insecurity significantly increased with older age, and the odds was lower for males when compared with non-males. Therefore, the study hypothesis is partially supported.

One of the strengths of the study is the large sample size making it possible to conduct robust data analysis. The study also provides one of the few findings on the impact of the COVID-19 pandemic on PLHIV in a country in sub-Saharan Africa; a region that is home to the majority of PLHIV [25].

Findings from the current study need to be considered in light of the following limitations. First, this is a cross-sectional study making it difficult to support causality. For example, it is not possible to determine the direction of the relationships of the factors associated with the outcome variables. Second, we also used self-reported measures of depression, anxiety and HIV status which are associated with high risk of social desirability bias. These forms of self-report may be more sensitive to identifying non-depressed, non-anxious and HIV negative individuals [36] because of the stigma associated with positive HIV and poor mental health status in sub-Saharan Africa [37,38,39,40]. Stigma may have resulted in under-reporting and an underestimation of the proportion of respondents who were depressed, anxious or PLHIV. There may also be the possibility of overestimating those who are HIV negative as a prior report from Nigeria noted that about 1.2% of adolescents living with HIV in the country report being HIV negative (this may or may or may not be applicable to adults) [35]. The sample is also a convenience sample with some associated challenges like the inability to generalise the result of the online survey to Nigeria as reflected by the skewness of the sample to those with tertiary education and possibly under-representation of those with low socioeconomic status; and a bias due to the exclusion of those who were able to fill the online questionnaire because of their access to the survey. This study may, however, be generalisable to the study population with tertiary education and may not be generalisable to settings with lower prevalence of HIV or a different profile of participants. The low values of the R2 of the models built for the logistic regression analysis supports our reflection that there might be other factors that may explain our study outcomes. The models, however, had good fits and outperformed the null models. In view of these limitations, the study is best considered exploratory and provides evidence to develop hypothesis for further studies.

Nevertheless, the study finding indicates that PLHIV are less prone to financial pressures, reductions in food intake and a negative impact of the epidemic on their daily life than do people not living with HIV. A plausible explanation may be that PLHIV lived a life of financial and food insecurity prior to the pandemic [41], and the magnitude of the new shock resulting from the pandemic may not have been as great for them. Also, PLHIV lives are majorly pre-occupied with addressing multiple vulnerabilities and thus, they have lived and adapted to a life of negative impacts prior to the pandemic [42,43]. It is therefore possible that during this pandemic, people not living with HIV have suffered more financial losses resulting from job loss [44], loss of or reduced wages [45] and investment losses [46,47]. These financial losses may have had a worse impact on food security among people not living with HIV, thereby having a larger impact on their daily life experiences. In effect, it appears that the life of PLHIV in Nigeria where less affected by the financial crisis, food crisis and other crises that may negatively impact the quality of life resulting from the pandemic through adaptation to a life of hardship [48] while these hardships are novel to people not living with HIV. Future studies need to explore the hypothesis posited here.

Financial insecurity is an emerging socioeconomic determinant of mental problems with indications that depression and anxiety are associated with higher risk of financial insecurity [49]. Similarly, financial and food insecurity increased anxiety and depression prior to [50,51] and during the COVID-19 pandemic [52]. In the study population, only the feeling of depression was associated with financial and food insecurity; and it resulted in lower odds of a decrease in food intake contrary to prior findings. Anxiety was not associated with either financial or food insecurity in this study. Past studies had indicated that anxiety and depression may cause either an increase or a decrease in food intake [53,54]; and that racial and ethnic differences in the prevalence and severity of anxiety and depression [55,56], there is no study accessible on racial and ethnic differences in the types of feeding disorder associated with anxiety and depression. This study may be a pointer to possible racial and ethnic differences in the way anxiety and depression affects food intake in different socio-cultural contexts.

Also, the finding that the negative impact of the COVID-19 on daily lives is less likely among those who felt anxious and depressed further complicates our findings. Prior studies had reported contrary findings: greater pandemic-related disruption in daily life were more likely in individuals with anxiety and depression symptoms [57]. Disruption of the daily lives of people by the pandemic also undermines the financial well-being of individuals and families [58], and this should circle back as a cause for anxiety or depression with a negative impact on food security and food intake. The impact of the pandemic on food intake, however, can vary between countries as there are a variety of COVID-19 related psychological changes that might have also affected food-related behaviours [59]. The reason for the findings in our study is unclear, though it does indicate a complex inter-relationship between these variables—mental health, financial security, decrease in food intake and perception of a negative impact of the pandemic on daily lives—that warrants further investigation. A plausibility is to explore the impact of culture and ethnicity as moderators of these relationships.

Men were less affected by financial insecurity, unlike prior studies that had indicated men were more affected by financial insecurity due to the perceived future associated risks [49]. The risk of other genders different from men facing higher financial insecurity due to the pandemic had been earlier reiterated. The informal economic sector, which was affected worse by the COVID-19 pandemic in Nigeria, is mainly run by women [60]. In the formal sector, women are more likely to be lower cadre staff, and thus are more likely to be laid off during the pandemic with the decrease in economic activities [60]. The entrenched role of women as care givers implies that they are less able to return fully to work during the pandemic when schools were closed as they had to provide care for the children [60]. the pandemic thereby further widened existing gender inequalities in the country. The risk of financial insecurity was also higher in the present study for older persons who are the major job holders in Nigeria (a lot of young persons do not have jobs [61,62]), thereby explaining why the study found older respondents were more likely to report financial insecurity.

## 5. Conclusions

We found a complex relationship between financial insecurity, decrease in food intake, mental health and HIV status and the negative daily life experiences of adults living in Nigeria during the first wave of the COVID-19 pandemic. Though some factors associated with the financial vulnerability and decrease in food intake of people in Nigeria during the pandemic were also associated with a negative impact of the COVID-19 pandemic on day-to-day life, the associations were not always in the same direction. Considering some of the limitations associated with the study design acknowledged here, further in-depth analyses to unravel these complexities may be warranted.

## Figures and Tables

**Table 1 ijerph-18-07925-t001:** Factors associated with financial insecurity, decrease in food intake and impact of the pandemic on daily life of adults living in Nigeria during the COVID-19 pandemic (*N* = 4439).

Variables	Total *n* (%)	Financial Insecurity			Decrease in Food Intake				Negative Impact of Pandemic on Daily Life		
		Yes 2487 *n* (%)	No 1454 *n* (%)	Not Sure 498 *n* (%)	*p* Value	Yes 907 *n* (%)	No 3532 *n* (%)	*p* Value	Yes 4029 *n* (%	No 410 *n* (%)	*p* Value
**Age in years** **Mean (SD)**	38.34 (11.6)	39.77 (12.0)	37.67 (11.3)	37.46 (11.9)	<0.001	38.37 (11.6)	38.33 (11.6)	0.934	38.43 (11.6)	37.45 (11.6)	0.676
**Sex at birth**											
Male	2076 (46.8)	1199 (57.8)	658 (31.7)	219 (10.5)	0.083	414 (19.9)	1662 (80.1)	0.448	1897 (91.4)	179 (8.6)	0.185
Female	2363 (53.2)	1288 (54.5)	796 (33.7)	279 (11.8)		493 (20.9)	1870 (79.1)		2132 (90.2)	231 (9.8)	
**Highest educational level**											
None	48 (1.1)	33 (68.8)	8 (16.7)	7 (14.5)	0.028	17 (35.4)	31 (64.6)	<0.001	45 (93.8)	3 (6.2)	0.002
Primary	84 (1.9)	44 (52.3)	26 (31.0)	14 (16.7)		40 (47.6)	44 (52.4)		83 (98.8)	1 (1.2)	
Secondary	715 (16.1)	422 (59.0)	208 (29.1)	85 (11.9)		163 (22.8)	552 (77.2)		628 (87.8)	87 (12.2)	
Tertiary	3592 (80.9)	1988 (55.4)	1212 (33.7)	392 (10.9)		687 (19.1)	2905 (80.9)		3273 (91.1)	319 (8.9)	
**Mental health status**											
*Anxiety*											
No	3693 (83.2)	2066 (55.9)	1230 (33.3)	397 (10.8)	0.041	718 (19.4)	2975 (80.6)	0.006	3289 (89.1)	404 (10.9)	<0.001
Yes	746 (16.8)	421 (56.5)	224 (30.0)	101 (13.5)		189 (25.3)	557 (74.7)		740 (99.2)	6 (0.8)	
*Depression*											
No	4050 (91.2)	2224 (54.9)	1376 (34.0)	450 (11.1)	<0.001	785 (19.4)	3265 (80.6)	<0.001	3653 (90.2)	397 (9.8)	<0.001
Yes	389 (8.8)	263 (67.6)	78 (20.1)	48 (12.3)		122 (31.4)	267 (68.6)		376 (96.7)	13 (3.3)	
**HIV infection status**											
Not living with HIV	3527 (79.5)	1923 (54.5)	1205 (34.2)	399 (11.3)	<0.001	626 (17.7)	2901 (82.3)	<0.001	3180 (90.2)	347 (9.8)	0.006
Living with HIV	912 (20.5)	564 (61.8)	249 (27.3)	99 (10.9)		281 (30.8)	631 (69.2)		849 (93.1)	63 (6.9)	

**Table 2 ijerph-18-07925-t002:** Logistic regression analysis showing the factors associated with financial security, decrease in food intake and impact of the pandemic on daily lives of adults living in Nigeria during the COVID-19 pandemic (*N* = 4439).

Variables	Financial Insecurity		Decrease in Food Intake		Negative Impact of Pandemic on Daily Life	
	AOR (95% CI)	*p* Value	AOR (95% CI)	*p* Value	AOR (95% CI)	*p* Value
**Age in years**	1.01 (1.00–1.02)	**<0.001**	-	-	-	-
**Sex at birth**						
Male (ref: not male)	0.84 (0.75–0.95)	0.005	1.005 (0.87–1.17)	0.944	0.82 (0.67–1.01)	0.060
**Highest educational level**						
None	1.00	-	1.00	-	1.00	-
Primary	2.06 (0.97–4.38)	0.060	0.59 (0.28–1.23)	0.159	0.18 (0.02–1.79)	0.143
Secondary	1.52 (0.80–2.88)	0.202	1.36 (0.72–2.55)	0.341	1.25 (0.37–4.25)	0.721
Tertiary	1.41 (0.75–2.66)	0.284	1.33 (0.72–2.48)	0.365	0.76 (0.22–2.57)	0.655
**Mental health status**						
*Anxiety*						
Yes (ref: No)	1.06 (0.90–1.26)	0.466	0.87 (0.72–1.06)	0.164	0.08 (0.03–0.17)	<0.001
*Depression*						
Yes (ref: No)	0.62 (0.50–0.79)	<0.001	0.62 (0.48–0.78)	<0.001	0.48 (0.27–0.85)	0.012
**HIV status**						
Living with HIV (ref: Not living with HIV)	0.70 (0.59–0.83)	<0.001	0.55 (0.46–0.67)	<0.001	0.68 (0.50–0.92)	0.012
**Nagelkerke R^2^**	0.020		0.037		0.070	
**Hosmer Lemeshow goodness of fit test**	13.73	0.089	3.58	0.612	8.07	0.152
**Omnibus test of model coefficients**	66.91	<0.001	105.73	<0.001	145.77	<0.001

## Data Availability

Data is available upon request from the corresponding author for this study.

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
