# Peer review of "Factors Associated with Financial Security, Food Security and Quality of Daily Lives of Residents in Nigeria during the First Wave of the COVID-19 Pandemic"

_ijerph, 2021, doi:10.3390/ijerph18157925_

Round 1

Reviewer 1 Report

The subject treated by the authors is of undoubted importance in terms of awareness for decision makers.

The statistical analysis is correct and consequent to the methodological choices made in the study design.

Unfortunately, the method of recruiting the sample, clearly dictated by the difficulties of the situation, exposes the conclusions to an extremely low level of reliability. The response variables defined as dichotomous, with the consequent choice of logistic regressions for the estimation of the adjusted ORs, produce a very high loss of information.

Another fundamental and critical aspect in the design is the fact that the response variables presumably correlate very much with each other and tend to describe the social condition of the subjects in a not particularly discriminating way.

The selection bias, although indicated in sections in the text, are not emphasized. In this context it is absolutely impossible to think of generalizing the results to the population but, at best, to a sub-group that is mostly privileged at least in terms of education. I believe that some of the interpretative difficulties derive from this selection bias.

I believe that the work can be published on condition of an explicit indication that it has exploratory purposes with an extensive and convincing discussion of limitations that are far from being "few". 

Author Response

Impact of the COVID-19 Pandemic on Financial Security, Food Security and Daily Living in Nigeria

ijerph-1272387

We thank the reviewers for the thorough edits to the manuscript. The feedbacks are kind, encouraging and constructive. We hope the responses have adequately addressed the issues raised by each reviewer. 

Reviewer 1

Comments and Suggestions for Authors

The subject treated by the authors is of undoubted importance in terms of awareness for decision makers.

RESPONSE: Thanks for the positive feedback

The statistical analysis is correct and consequent to the methodological choices made in the study design.

RESPONSE: Thanks for the positive feedback

Unfortunately, the method of recruiting the sample, clearly dictated by the difficulties of the situation, exposes the conclusions to an extremely low level of reliability.

RESPONSE: We have identified this as a study limitation and included it in the study limitation section. We wrote: The sample is also a convenient sample with some associated challenges like the inability to generalise the result of the survey to Nigeria as reflected by the skewness of the sample to those with tertiary education and possibly under-representation of those with low socioeconomic status; and a bias due to the exclusion of those who were able to fill the questionnaire because of their access to the survey. This study may, however, be generalisable to the study population with tertiary education.

The response variables defined as dichotomous, with the consequent choice of logistic regressions for the estimation of the adjusted ORs, produce a very high loss of information.

RESPONSE: Thanks for raising this. It is correct that categorizing quantitative variables into yes/ no levels may lead to loss of sensitivity that may reflect on the power to detect significant associations. The first response variable (financial insecurity) was included in the survey as is, a yes/ no variable based on previous studies. In the second variable (decrease in food intake), we combined two categories into one because we were interested on decrease in food intake since the pandemic is assumed to affect people’s ability to get their food. In the third variable (negative impact of the pandemic on quality of life), we combined several categories to obtain the binary response aiming at balanced categories since the original subgroups had a concentration of responses in one or two categories leaving other categories with very few responses. After these categorizations, there were still significant associations indicating that the general pattern of associations was not missed and combining categories did not affect the ability to detect associations.

Another fundamental and critical aspect in the design is the fact that the response variables presumably correlate very much with each other and tend to describe the social condition of the subjects in a not particularly discriminating way.

RESPONSE: We checked the variables and had eliminated variables that showed co-linearity when analysing the data. We agree about the possible complexity of the relationships between the variables. Yet, each variable is independent. This study did not get into these details. Future studies should be able to unravel the role of the different factors and how they are jointly affected by the independent factors.

The selection bias, although indicated in sections in the text, are not emphasized. In this context it is absolutely impossible to think of generalizing the results to the population but, at best, to a sub-group that is mostly privileged at least in terms of education. I believe that some of the interpretative difficulties derive from this selection bias.

RESPONSE: Thanks again for highlighting this. We have noted this possibility in the study limitation also.

I believe that the work can be published on condition of an explicit indication that it has exploratory purposes with an extensive and convincing discussion of limitations that are far from being "few". 

RESPONSE: we have identified the limitation are not few. We have further highlighted the limitations they are and the potential use of the study – for exploratory purposes. We wrote: In view of these limitations, the study is best considered exploratory; and provides evidence to develop hypothesis for further studies.

Reviewer 2 Report

The study aims to identify factors associated with financial loss, food insecurity and impact on daily lives in Nigeria during the first wave of the COVID-19 pandemic. The study have significant public health implications and I would love to see articles with such topic being published. However, the article needs to be revised to better define the research question, clarify method section and provide more discussions. I hope the following are useful considerations. 

  1. The title is "Impact of the COVID-19" while the aim in abstract is "to identify factors... during COVID-19". The title gives me the impression that the authors would compare situation before and during COVID-19. Please consider re-wording the title and align with study aims and analyses.
  2. I don't quite understand the authors's statement in line 143-144. Depression and anxiety may be mediators between which exposure and which outcome? If anxiety and depression are the mediators, do the authors consider potential mediator analyses? 
  3. Why the study population is sampled from Glocal while to focus of this article is to identify factors associated with financial and food insecurity, and the impacts on the daily lives of adults in Nigeria during the first wave of the COVID-19 pandemic. Nigeria is home to second largest population of PLHIV, however, if we sample data globally, would HIV be a key factor driving financial loss , food intake and daily lives during COVID-19?
  4. So the actual analysis population is for respondents from Nigeria? How did the authors identify participants from Nigeria. Please clarify in method section. 

    As the authors mentioned in introduction section, people living in Nigeria tend to have limited access to Internet. Is the data collected using online survey representative of the Nigeria population? 

  5. I wonder what the results would be if the study is conducted before COVID-19? Will people living with HIV still be less prone to financial loss in that period? 
    Is there any other studies that focus on HIV population and their life/financial situation prior to COVID-19? What are the study results? 

Author Response

Impact of the COVID-19 Pandemic on Financial Security, Food Security and Daily Living in Nigeria

ijerph-1272387

We thank the reviewers for the thorough edits to the manuscript. The feedback are kind, encouraging and constructive. We hope the responses have adequately addressed the issues raised by each reviewer. 

Reviewer 2

Comments and Suggestions for Authors

The study aims to identify factors associated with financial loss, food insecurity and impact on daily lives in Nigeria during the first wave of the COVID-19 pandemic. The study have significant public health implications and I would love to see articles with such topic being published. However, the article needs to be revised to better define the research question, clarify method section and provide more discussions. I hope the following are useful considerations. 

RESPONSE: Thanks for the positive feedback

  1. The title is "Impact of the COVID-19" while the aim in abstract is "to identify factors... during COVID-19". The title gives me the impression that the authors would compare situation before and during COVID-19. Please consider re-wording the title and align with study aims and analyses.

RESPONSE: Thanks for this suggestion. We have revised the title to: Factors associated with financial security, food security and quality of daily lives of residents in Nigeria during the first wave of the COVID-19 pandemic

  1. I don't quite understand the authors's statement in line 143-144. Depression and anxiety may be mediators between which exposure and which outcome? If anxiety and depression are the mediators, do the authors consider potential mediator analyses? 

RESPONSE:  Thanks for identifying this confusing statement. We have revised the statement to address the issues raised in the discussion more appropriately. We wrote: Depression and anxiety may also affect the use of telemedicine services thereby affecting daily lives due to poor access to healthcare during the pandemic.

  1. Why the study population is sampled from Glocal while to focus of this article is to identify factors associated with financial and food insecurity, and the impacts on the daily lives of adults in Nigeria during the first wave of the COVID-19 pandemic. Nigeria is home to second largest population of PLHIV, however, if we sample data globally, would HIV be a key factor driving financial loss , food intake and daily lives during COVID-19?

RESPONSE: We collected data on a global level from several countries and used the dataset for this and other studies. Other studies- yet under review- used the entire dataset. In this MS, we focused on Nigeria to study the characteristic factors including HIV status. We think the decision to do an analysis of a subset of sample of the global data is appropriate. The sample for this study were respondents who specifically identified they were resident in Nigeria during the pandemic. We added a part in the Discussion about this..

  1. So the actual analysis population is for respondents from Nigeria? How did the authors identify participants from Nigeria. Please clarify in method section. 

RESPONSE: All participants were required to identify the country of residence at the time of filling the questionnaire which was within the period of the first wave of the pandemic. We only curated the data of respondents who identified they were resident in Nigeria. The title of the manuscript also notes this.

As the authors mentioned in introduction section, people living in Nigeria tend to have limited access to Internet. Is the data collected using online survey representative of the Nigeria population? 

RESPONSE: we had identified this limitation including the inability to generalise the survey to all Nigerians. We hope the detailed discussion about the limitation of the study addresses the concern of the reviewer.

  1. I wonder what the results would be if the study is conducted before COVID-19? Will people living with HIV still be less prone to financial loss in that period? 

RESPONSE: The data was collected during the first wave of COVID-19 in Nigeria. We think our finding will represent the period in time that the reviewer refers too. We however, acknowledge that cross-sectional studies have their limitations as life situation changes for many reasons. We have also acknowledged this limitation.

Is there any other studies that focus on HIV population and their life/financial situation prior to COVID-19? What are the study results? 

RESPONSE: In the Introduction, we presented a brief overview about the findings of previous studies (references #11 to 15) showing the PLHIV have several challenges even before the COVID-19 pandemic. 

Round 2

Reviewer 1 Report

The authors provided significant efforts to manage my concerns. I think the manuscript can be published

Reviewer 2 Report

All my comments are appropriately addressed by the authors.

This manuscript is a resubmission of an earlier submission. The following is a list of the peer review reports and author responses from that submission.